# Superior Performance with Diversified Strategic Control in FPS games Using General Reinforcement Learning

## Abstract

This paper offers an overall solution for first-person shooter (FPS) games to achieve superior performance using general reinforcement learning (RL). We introduce an agent in ViZDoom that can surpass previous top agents ranked in the open ViZDoom AI Competitions by a large margin. The proposed framework consists of a number of generally applicable techniques, including hindsight experience replay (HER) based navigation, hindsight proximal policy optimization (HPPO), rule-guided policy search (RGPS), prioritized fictitious self-play (PFSP), and diversified strategic control (DSC). The proposed agent outperforms existing agents by taking advantage of diversified and human-like strategies, instead of larger neural networks, more accurate frag skills, or hand-craft tricks, etc. We provide comprehensive analysis and experiments to elaborate the effect of each component in affecting the agent performance, and demonstrate that the proposed and adopted techniques are important to achieve superior performance in general end-to-end FPS games. The proposed methods can contribute to other games and real-world tasks which also require spatial navigation and diversified behaviors.

## 1 Introduction

Games have been considered as challenging benchmarks in evaluating reinforcement learning (RL) algorithms. In games, agent can feel free to explore through infinite trial and error. OpenAI Gym (Brockman et al., 2016) has provided many wrapped game environments, such as Atari games (Bellemare et al., 2013) and Mujuco robotic control problems (Todorov et al., 2012), through an unified interface. Meanwhile, many RL algorithms, such as TRPO (Schulman et al., 2015), PPO (Schulman et al., 2017), DQN (Mnih et al., 2015) and DDPG (Lillicrap et al., 2019), etc., have been demonstrated to achieve superior performance in these environments. Recent advances in solving more complex games, including GO (Silver et al., 2016), DOTA 2 (Berner et al., 2019), and StarCraft II (Vinyals et al., 2019), further demonstrate that general RL method can be widely applied in solving simulated environments.

First-person shooter (FPS) games, such as Quake (Jaderberg et al., 2019) and Doom (Kempka et al., 2016), are also known by their complexity, while they are considered to be closer to real-world tasks because solving FPS games requires perfect navigation skills conditioning on raw screen/camera input. Normally, the partially observed screen only contains limited information, and the agent cannot obtain the global state of the environment and the information of other agents. Many real-life applications, such as searching and rescuing robotics, and autonomous driving, etc., are very similar to FPS games. Another challenge posed in these environments is that the state transition is not static. However, most existing intelligent agents are specifically trained to work well in some fixed environments, while they are unable to act diversely when the environment dynamics changes.

In this paper, we take FPS games as the benchmark environment. We propose an overall solution to train intelligent agent in FPS games that can dynamically adjust its strategy according to environmental changes induced by diverse opponents. Specifically, we focus on the game of ViZDoom. We introduce an agent with diversified strategic control (DSC) that can surpass previous top agents

ranked in the open ViZDoom AI Competitions [1,2] by a large margin. The solution framework proposed in this paper consists of a number of general techniques in RL, including hindsight experience replay (HER) (Andrychowicz et al., 2017), hindsight proximal policy optimization (HPPO), rule-guided policy search (RGPS) (Han et al., 2021), prioritized fictitious self-play (PFSP) (Vinyals et al., 2019), and diversified strategic control (DSC), etc. We conduct comprehensive experimental results to show the importance of each of the introduced techniques in training intelligent agents. Our proposed methods can contribute to other games and real-world tasks which require spatial navigation and diversified behaviors that are similar to FPS games.

## 2 RELATED WORK

The proposed learning framework is based on the widely adopted actor-critic RL methods, in which a value function and a policy are learned simultaneously. Many algorithms, such as TRPO (Schulman et al., 2015), PPO (Schulman et al., 2017), DDPG (Lillicrap et al., 2019) and TD3 (Fujimoto et al., 2018), etc., all follow the actor-critic architecture and have achieved state-of-the-art results in various applications. In our learning framework, we will adopt PPO as a baseline RL algorithm.

In ViZDoom, a most fundamental skill that the agent has to learn is navigation. Considering a navigation task, we need to define a target in advance. Therefore, the navigation problem in ViZDoom can naturally be viewed as a goal-conditioned task, which has been well-studied in the RL literature. Among many advanced algorithms, hindsight experience replay (HER) (Andrychowicz et al., 2017) provides a smart solution that it replaces the goal in failed experiences with any practically achieved one to pretend that the agent obtains a positive reward. By doing so, the agent has a much higher chance to see successful trajectories. There have been rich approaches demonstrating that HER and its variants are effective in solving goal-conditioned tasks.

In complex tasks like ViZDoom, the problem can usually be decomposed into multiple stages. As we will show in the method section, our learning framework consists of three training stages, each of which focuses on solving a specific problem. One challenge in multi-stage training is that the agent at a later stage is easy to forget what has been learned in early stages. To avoid this embarrassing situation, policy distillation (Rusu et al., 2015) has been imported to keep the training policy staying close to the parameters trained in earlier stages. For example, in (Vinyals et al., 2019), the training policy in RL phase is kept close to the intial policy trained using supervised learning via a policy distillation term. Our learning framework also utilize policy distillation as an important component.

To obtain intelligent and superior AI agents in competitive multi-agent games, self-play (SP) (Silver et al., 2018) is often necessary to generate high-quality competitions. It conducts an automatic curriculum learning by letting the agent combat with itself or its own historical models. In addition to AlphaGo (Silver et al., 2016), SP has been demonstrated effective in many other complex games, such as hide-and-seek environment (Baker et al., 2019) and DOTA 2 (Berner et al., 2019). Variants of SP, such prioritized fictitious self-play (PFSP) proposed in AlphaStar (Vinyals et al., 2019), have also been verified to be effective. We will use SP and PFSP in our learning framework to train intelligent agents.

## 3 BACKGROUND IN VIZDOOM

ViZDoom (Kempka et al., 2016) is a complex FPS game, and its formal competition scheme is using the Deathmatch mode, in which all agents fight against each other. In this mode, agent with the highest Frag score, which is defined as defeating counts minus the suicide counts, wins the game. Unlike the games of Go, Atari, and Starcraft II, where the player only needs to interact with the environment or compete with another opponent in one versus one zero-sum game, in the Deathmatch of ViZDoom Competitions in 2016 and 2017, the player needs to fight against other 7 independent agents with diverse strategies.

ViZDoom is a multi-agent environment with imperfect information that each agent can only observes very limited information in the environment at a time step. In addition, the agent takes raw image

---

[1]http://vizdoom.cs.put.edu.pl/competitions/vdaic-2016-cig
[2]http://vizdoom.cs.put.edu.pl/competitions/vdaic-2017-cig

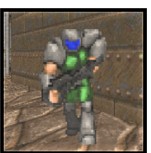

**Player**
10 action button + 4 angle button.
button independent.
Free combination.

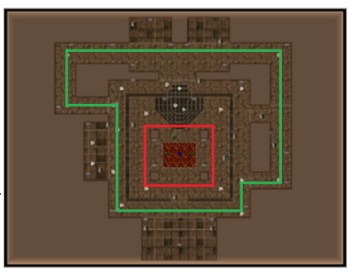

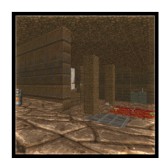

**Map**
Corridors, internal and external
platforms have different heights.
The connection between the inside
and the outside is a staircase and
three windows.

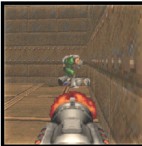

**Score**
Defeat other agents to get points

**Rules**
Eight players play against each other.
Rank based on the number of enemies
defeated minus the number of suicides.
Randomly reborn after ten seconds.

Figure 1: The ViZDoom environment.

as input and we need to train end-to-end policy, where the state space is of large image size. These characteristics of ViZDoom pose great challenges to training the agent from scratch.

There are already some works that have achieved well performance in ViZDoom. Clyde (Ratcliffe et al., 2017) imports the LSTM (Hochreiter & Schmidhuber, 1997) structure to the Asynchronous Advantage Actor-Critic (A3C) algorithm (Mnih et al., 2016), resulting a basic intelligent agent in ViZDoom. F1 (Wu & Tian, 2016) who won the championship of ViZDoom AI Competitions 2016 is also trained with A3C algorithm. It also uses human-prior knowledge to conduct reward shaping and curriculum design to assist the agent learning. There is also another work (Huang et al., 2019) using deep recurrent Q-learning network as a high-level controller. It combines auxiliary tasks (opponent detection and depth prediction) to manage the combo-actions. However, these agents all have a common shortcoming: the strategy of their agents is fixed. For example, in the map (see Figure. 1) of ViZDoom, one agent prefers to circle around the outer loop (green line), and the other agent prefers to act in the middle magma area (red line). These two agents may never meet each other in a competition game, and thus can not get Frag scores either. Therefore, how to obtain diversified strategy in ViZDoom still remains an open problem in the literature.

## 4 METHODOLOGY

The entire learning framework consists of three stages of training, by decomposing the complex task in the ViZDoom game into different levels of control, i.e., navigation, frag and strategic control. For complex systems, multi-stage learning is necessary and more efficient to build strong artificial intelligence. An overview of our multi-stage learning framework is shown in Figure 2.

We briefly explain the framework here and elaborate the details in the following subsections. In stage 1, we aim to obtain an expert navigation agent, which only takes moving actions. Specifically, we enable the agent to navigate by following a given target, and this is very useful for stage 3 when training the agent at a strategic level. Apparently, navigation to a given target is the essential problem considered by goal-conditioned RL (Andrychowicz et al., 2017). It has been well demonstrated that RL methods using hindsight experience replay (HER) (Andrychowicz et al., 2017) are effective in solving goal-conditioned tasks. Therefore, we adopt HER in our navigation stage. We are not aware of any game AIs that explicitly take advantage of HER. The original HER method is implemented with the off policy method such as DDPG and TD3. For training strong AIs in complex video games, it has been demonstrated that actor-critic algorithms, such as PPO, with discrete actions are more effective (Schulman et al., 2017). In order to apply PPO and HER together, we adopt the same idea as the recent Hindsight Trust Region Policy Optimization (HTRPO) method (Zhang et al., 2021), proposed the Hindsight Proximal Policy Optimization method, which we call HPPO. Based on an excellent navigation policy, in the second training stage, we let the agent participate in the formal battle game in ViZDoom, which is also termed as 'Frag' in ViZDoom. At this stage, we use Prioritized Fictitious Self-Play (PFSP) (Vinyals et al., 2019) to let the training agent play against itself and its historical versions with prioritization. To conveniently reuse the navigation policy, we

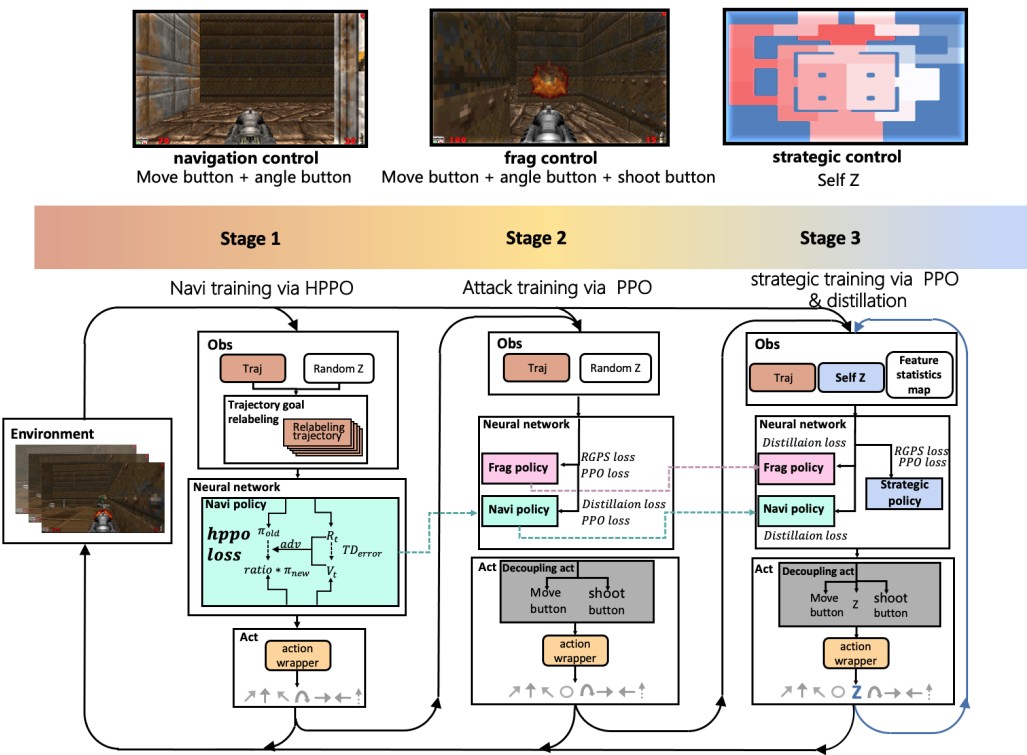

Figure 2: The infrastructure of our multi-stage learning system. Among the stages, the dashed lines represent model reuse. In our implementation, the goals are encoded into high-dimensional one-hot vectors, which are denoted as the variable $Z$.

create a new action head for frag in addition to the moving actions. So far, the trained agent can play formal ViZDoom games by randomly feeding a navigation target to the policy. In a formal ViZDoom match in the historical competitions, one player will play against another 7 players in the map. Therefore, in the last stage, we propose to learn a policy at a strategic level to output a specific target, so that the agent itself can decide where to navigate. Again, to smoothly reuse the pre-trained policy in the previous two stages, we create a new action head for deciding the target. In this way, the agent should be aware of its opponents' strategies and then decide where to go to encounter them. A detailed infrastructure of the proposed multi-stage learning system has been depicted in Figure 2.

### 4.1 STAGE 1: GOAL-CONDITIONED NAVIGATION

FPS games pose great challenges on the player's moving and shooting skills. When human players play FPS games, they always demonstrate diverse strategies for moving and shooting, conditioning on their opponents' strategies in one episode. For example, if a professional player knows that one opponent is used to hide at a specific location in the map, he/she will navigate to somewhere that can easily monitor or shoot that location. For AI agents, an explicit way to modeling such navigation strategies is to define goals. Then, the problem is naturally converted to the well-studied goal-conditioned RL. There have been a rich literature on goal-condition RL methods, among which one of the most effective one is using hindsight experience replay (HER). Researches based on HER often consider continuous control tasks based on DDPG, and these tasks are relatively simple environments. In ViZDoom, the action space is discrete, and we choose to apply PPO and HER to solve the navigation problem. Directly training PPO using hindsight experiences is problematic, since PPO is an on-policy method while hindsight experiences are handcraft data which does not follow the policy any more. Therefore, we have to correct the objective in PPO by re-sampling trajectories from the hindsight experiences, following the Hindsight TRPO method (Zhang et al., 2019). This method is referred to as the Hindsight PPO (HPPO). Formally, HPPO maximizes the

following objective:

$$L_{\theta_{old}}(\theta) = \mathop{\mathbb{E}}_{g',\tau} \sum_{t=0}^{\infty} \left[ \min \left( \prod_{k=0}^{t} \frac{\pi_{\theta_{old}}(a_k \mid s_k, g')}{\pi_{\theta_{old}}(a_k \mid s_k, g)} \gamma^t \frac{\pi_{\theta}(a_t \mid s_t, g')}{\pi_{\theta_{old}}(a_t \mid s_t, g')} A^{\pi_{\theta_{old}}}(s_t, a_t, g') \right. \right.$$
$$\left. \left. \text{clip} \left( \prod_{k=0}^{t} \frac{\pi_{\theta_{old}}(a_k \mid s_k, g')}{\pi_{\theta_{old}}(a_k \mid s_k, g)} \gamma^t \frac{\pi_{\theta}(a_t \mid s_t, g')}{\pi_{\theta_{old}}(a_t \mid s_t, g')}, 1 - \varepsilon, 1 + \varepsilon \right) A^{\pi_{\theta_{old}}}(s_t, a_t, g') \right) \right],$$

(1)

where $g$ and $g'$ denote the original and hindsight goals, respectively, and $\tau$ indicates the trajectory $(s_0, a_0, r_0, g, \cdots)$. $\theta$ is the policy parameter and $\theta_{old}$ is the parameter since last update. $A^{\pi_{\theta_{old}}}(s_t, a_t, g')$ is the advantage function. $\varepsilon$ is the clip range and we set $\varepsilon = 0.2$ as suggested by PPO. When $g = g'$, the object reduces to the original PPO objective.

In ViZDoom, we discretize the map into 20 areas (shown in Figure 8), whose center positions compose the set of goals. We simply code each goal as a 20-dimensional one-hot embedding, while it is clear that the goals can be defined in other ways.

Another practically useful technique is to transfer a sequence of consecutive actions into more efficient action combos assisting moving. This module is referred to as an action wrapper. For example, when the action wrapper detects 4 consecutive discrete actions of turning to left (by one degree), the action wrapper will output a immediate action of turning left by 20 degrees. In addition, when the policy network continues outputting more than 4 moving forward commands, the action wrapper will push a speedup moving forward command. These action combos can effectively help the agent to perform human-like behaviors.

### 4.2 STAGE 2: FRAG BY MAINTAINING THE EXPERTISE IN MULTI-GOAL NAVIGATION

At the second stage, we allow the agent to play formal full game in ViZDoom competitions, inheriting the expertise in multi-goal navigation learned from stage 1. To conveniently reuse the trained parameters from stage 1, we create a separate action head for controlling frag, with only a few parameters that are randomly initialized. The head outputs a binary value indicating whether the agent shoots or not at the current step. Note that in ViZDoom, the player is able to perform navigation and frag at the same time step. By decoupling the navigation actions and frag actions, the agent would explore better policy to coordinate navigation and frag. This is similar to control multi-agent systems with a centralized policy.

To promote data efficiency, we take advantage of the previously demonstrated effective technique in StarCraft II, named Rule-Guided Policy Search (RGPS) (Han et al., 2021). RGPS distills some straightforward domain knowledge into a pure neural network to skip unnecessary explorations. For training the frag head, we use a simple handcraft rule that if an opponent shows up in a certain range of the center of the raw screen, then shoot, without executing any moving actions.

Moreover, to make sure that the policy network avoid forgetting the navigation skills learned at the first stage, we maintain a policy distillation term to let the policy trained in the second stage stay close to that trained in the first stage only for the navigation head. It's important to emphasize that, the reason why we continue to train the navigation head is that the navigation head is not just useful for moving, it is also important to coordinate with the frag action for accurate movement and shooting. The overall objective function at this stage is:

$$L_{\text{Frag}} = L_{\text{PPO}}^{\text{frag+navigation}} + \lambda L_{\text{RGPS}}^{\text{frag}} + \mu L_{\text{distill}}^{\text{navigation}},$$

(2)

where $L_{\text{PPO}}$ indicates the standard PPO loss for optimizing both the navigation and frag policies, $L_{\text{RGPS}}$ is the RGPS term specially defined for the frag head, and $L_{\text{distill}}$ is a policy distillation term preventing navigation performance decrease for the navigation head. $\lambda$ and $\mu$ are two hyperparameters.

### 4.3 STAGE 3: DIVERSIFIED STRATEGIC CONTROL

An important challenge in FPS games sharing the same competition mode in ViZDoom is that in one match, there are multiple diverse players, e.g., 8 in ViZDoom, combating with each other. In the game of GO (Silver et al., 2016), DOTA 2 (Berner et al., 2019) and StarCraft II (Vinyals et al.,

2019), the competitions are between two players, i.e., one versus one zero-sum games. Therefore, simultaneously combating with many opponents requires more diversified skills from strategic level.

At the final training stage, we aim to train a strategic policy that can dynamically control the agent's strategy conditioning on the observed opponents' behaviors. Specifically, we let the agent decide the current goal by itself. This is achieved by creating a new action head again and reuse the previous trained parameters from stage 2. The strategic policy outputs a goal selected from the goal set, and then the goal is fed into the navigation and frag heads to perform navigation and frag actions. Such auto-regressive actions have been investigated in AlphaStar (Vinyals et al., 2019) as well. Different from stage 2, we abandon PFSP and propose a new game matching scheme to generate self-play matches. In a formal game of ViZDoom, there are 8 players in total to compete with each other. For a specific episode, we let the 7 opponents execute a fixed strategy randomly sampled from the beginning of the match. This can be easily achieved by sampling a fixed goal for them. In addition to the observation taken by the navigation and frag heads, the strategic policy takes as input an additional feature, that is a statistical map calculated from the cumulative historical imperfect observation in this episode. Specifically, the map records a score for each pixel, where the score is the cumulative Frag scores the agent gains at this specific pixel. At the testing phase in ViZDoom, the raw observation is the screen image, and a bird view of the entire map by accurately positioning the agent is not available. Therefore, we train an additional auxiliary network to positioning the agent in the bird view map by taking the raw screen image as input. This auxiliary network is trained using supervised learning during the RL training phase, by cheating on acquiring the current global coordinates of the agent.

It should be emphasized that the strategic policy should control the agent infrequently. Otherwise, the agent might change its goal to violate its current strategy very often. To void the situation, the strategic action head is activated only when the agent successfully reaches a goal or reborn. This poses a new challenge that the amount of effective data samples collected for training the strategic policy will be considerably limited. For example, there are around 10,000 moves in a formal match, but the number of frames at which the strategic head is activated is only about 30-40. Therefore, to enhance the exploration efficiency at the strategic level, we employ the RGPS method (Han et al., 2021) to incorporate some straightforward domain knowledge to guide strategic exploration. The handcraft principle is quite simple that the probability of selecting a goal is proportional to the summation of the recorded Frag scores belonging to the according area in the statistical map (see Figure 8). The objective function at this stage is

$$L_{\text{Strategic}} = L_{\text{PPO}}^{\text{strategic}} + \lambda L_{\text{RGPS}}^{\text{strategic}}, \tag{3}$$

where both the PPO loss and RGPS loss are defined on the strategic policy, and the navigation and frag heads are not updated any more in this stage.

## 5 EXPERIMENTS

In this section, we provide comprehensive empirical studies for the proposed learning framework. The finally trained agent is referred to as the Diversified Strategic Control (DSC) Agent. In Section 5.1, we visualize the representation learned by the last embedding layer shared by the policy and value networks to provide in-depth understanding of the agent. In Section 5.2, we report various ablation experiments to study the impact of the proposed components in affecting the agent performance. Finally, we report the overall testing performance of DSC-Agent by creating a competition league, composed by the top ranking agents in previous open ViZDoom AI competitions, including

- Marvin (Wydmuch et al., 2018), the champion in Track1 of ViZDoom AI Competition 2017, which is trained by first imitating human expert replays and then by RL.

- F1 (Wu & Tian, 2016), the champion in Track1 of ViZDoom AI Competition 2016, which is trained with curriculum reinforcement learning.

- Axon (Wydmuch et al., 2018), the third place in Track1 of ViZDoom AI Competition 2017, which uses a similar training pipeline as Marvin.

- YanShi (Wydmuch et al., 2018), the sixth place in Track1 of ViZDoom AI Competition 2017, which adopts a perception module and a planning module.

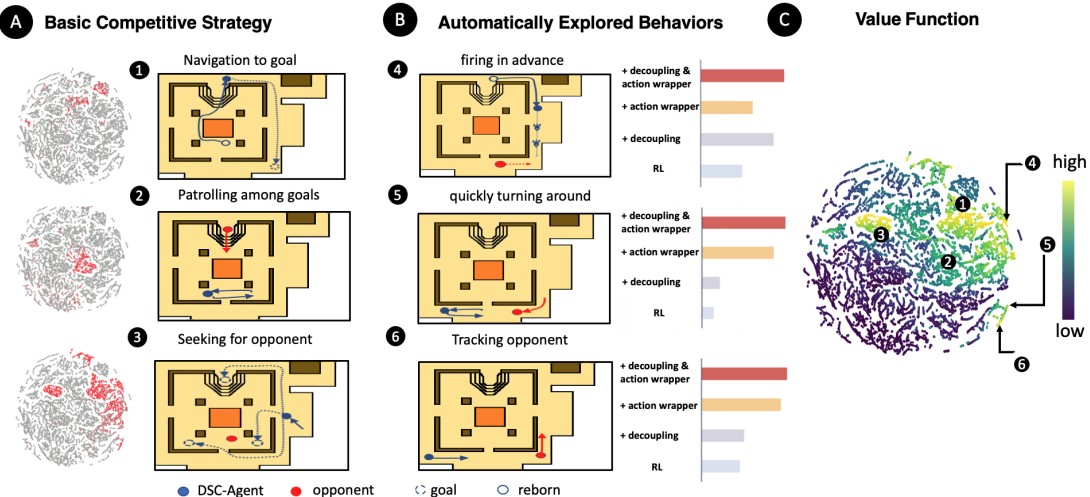

Figure 3: (A) Visualization of the neuron activation for three basic competitive strategies learned by DSC-Agent. (B) Three novel high-level tactics discovered by DSC-Agent, with their statistics of occurrence per match for different configurations. (C) Visualization of the value function predicted by DSC-Agent.

## 5.1 VISUALIZATION OF THE LEARNED REPRESENTATION

We use t-SNE (Van der Maaten & Hinton, 2008) to visualize the learned representations in the last embedding layer of the agent's network. We evaluate the DSC-Agent by performing $10^6$ time steps in a formal match, and use the generated data for visualization by t-SNE. Figure 3 shows the results. In Figure 3(A), t-SNE visualizes the activated neurons according to three frequently observed behavior patterns at the strategic level: (1) navigation to the target, (2) patrolling among targets, and (3) seeking for opponents. The visualization illustrates the learned representations memorize these strategic behaviors via different neuron regions, similar to human brains. These three strategic behaviors are as expected, since the proposed multi-stage learning framework is carefully developed to induce these behaviors. However, surprisingly, Figure 3(B) shows that three additional behavior types are automatically discovered by the agent: (4) firing in advance, (5) quickly turning around to face the opponent, and (6) tracking an observed opponent. Behaviors (4)-(6) are observed less frequently compared to (1)-(3), and they are scattered on the neurons as denoted in the visualized value function in Figure 3(C). On the right-hand side of Figure 3(B), the counts of observed behaviors (4)-(6) are recorded for different configurations, and we can observe that decoupling the action into multiple heads and using action wrapper are helpful to discover these novel tactics. Figure 3(C) visualizes the value of the corresponding behaviors, and we can observe that all the previously learned behaviors deserve much higher values in this neuron map.

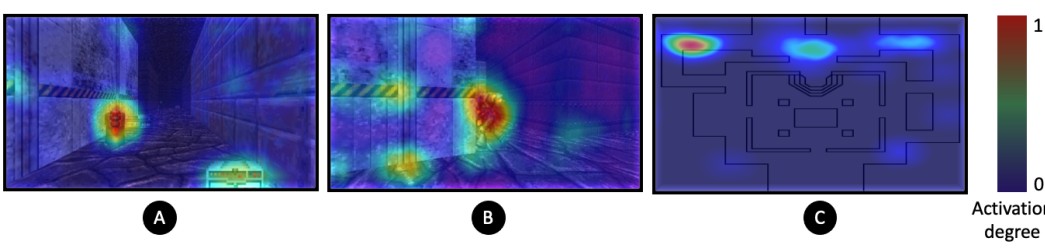

Figure 4: (A) and (B) Visual representation of DSC-Agent's image processing network by using Class Activation Mapping(CAM). (C) Visual representation of DSC-Agent's strategic control processing network by using Class Activation Mapping(CAM).

We also visualize where DSC-Agent pays more attention in the raw image input by using Class Activation Mapping (CAM) (Zhou et al., 2016). Figure 4(A) and (B) show the network activation for each pixel on the input screen when observing supplies and opponents, respectively. These figures demonstrate that the agent can accurately focus on these important objects.

In Figure 4(C), we also visualize the attention on the bird view map constructed by the auxiliary network mentioned in Section 4.1. It demonstrates that the agent focuses on a few important locations from the strategic level.

## 5.2 EVALUATION ON STRATEGIC DIVERSITY

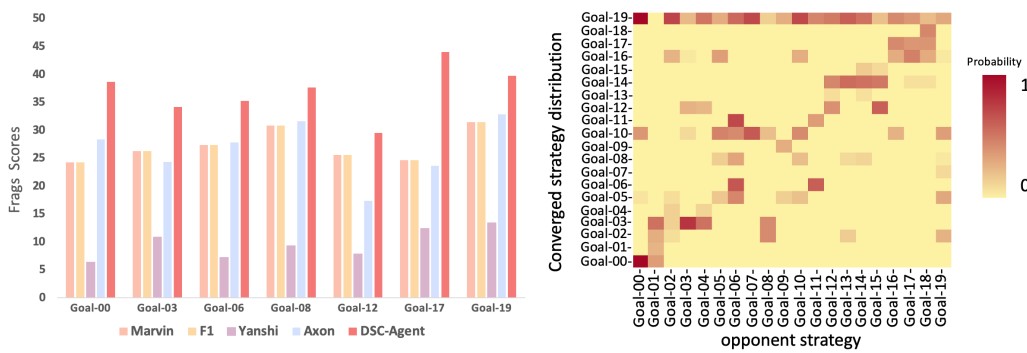

Figure 5: Left: average Frags scores for each compared AI agent when playing against different testing opponents. The scores are computed by averaging over 10 episodes, each of which lasts for 10 minutes. Right: a matrix showing the output distribution of the strategic policy in DSC-Agent (rows) against diverse opponents (columns).

As we have claimed multiple times that DSC-Agent is superior compared to existing state-of-the-art AIs by deploying a strategic policy that can decide the goal for itself. To verify this, we perform evaluation on strategic diversity in this section.

Recall that the agent trained at stage 2 can take any goal as input and act according to the goal in full ViZDoom match. Therefore, it is naturally to create a set of testing agents by selecting several representative goals from the goal set and feeding them into the trained agent at stage 2.

The left figure in Figure 5 shows the testing scores for all the state-of-the-art methods. Each bar indicates the average testing Frags score of a specific AI agent against 7 DSC-Agents from stage 2 by feeding a specific goal. As we can observe, when playing against opponents with different strategies, the compared AIs also show unstable scores. It is obvious that DSC-Agent can achieve the highest scores against all the opponents. The right figure in Figure 5 shows a matrix, in which the row indicates the output distribution of the strategic policy in DSC-Agent over the goals and the column indicates the strategy of the opponents. By referencing the goal map in Figure 8, it is easy to see that the goals selected by DSC-Agent in rows are closely related to the goal indicated in the column, spatially. For example, for the opponent taking Goal-00 strategy, DSC-Agent prefers to take strategies with Goal-00, Goal-10 and Goal-19, from which locations the agent can directly observe the location indicated by Goal-00.

## 5.3 ABLATION STUDIES

We provide comprehensive ablation studies to verify the importance of each component proposed in our learning framework. In Figure 6(A), we evaluate the navigation performance in stage 1 by comparing different configurations with PPO, HPPO and action wrapper. The results demonstrated that HPPO is much more efficient than PPO for goal-conditioned navigation, and its performance is further enhanced by applying the action wrapper. Figure 6(B) reports the final performance of DSC-Agent by testing against other state-of-the-art AIs in Figure 7, by varying the game matching scheme in stage 2. Similar to the results in (Vinyals et al., 2019), a combination of PFSP and SP is the most effective way. Similarly, Figure 6(C) shows the importance of action decoupling, RGPS in

stages 2 and 3, and strategic control. As we can observe, all the components are important to allow DSC-Agent to achieve the final performance.

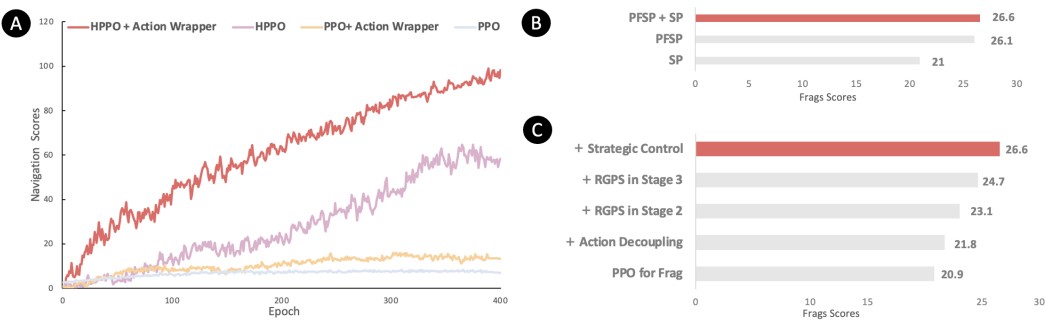

Figure 6: (A) Comparison of HPPO, PPO and action wrapper in stage 1. (B) Evaluating the game matching scheme used in stage 2 on affecting the final testing performance of DSC-Agent. (C) Evaluation of the importance of action decoupling, RGPS in stages 2 and 3, and strategic control. All scores are averaged over 10 episodes, each of which has 10 minutes game length, during the Deathmatch in Figure 7.

## 5.4 FINAL COMPETITION RESULTS

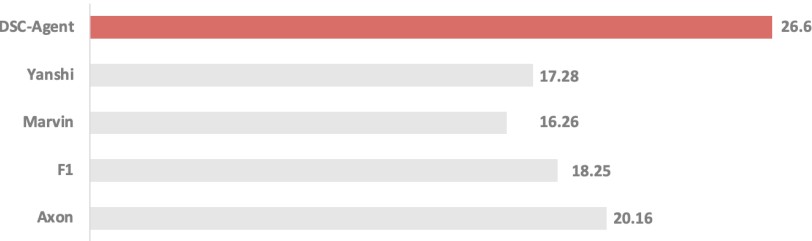

Figure 7: DSC-Agent vs. previous top AIs in Deathmatch.

Finally, we report the most important results by creating a formal league competition among Marvin, F1, Axon, YanShi, and DSC-Agent. The regulation of the competition follows that in Track1 of ViZDoom AI Competition 2017. We use a known map and fixed weapons in Limited Deathmatch. All the AI agents play against each other for 12 rounds of 10 minutes Deathmatch. All the AI agents are implemented using a Tesla M40 GPU, and the Frag scores are calculated as number of frags minus the suicide penalty. The overall results are shown in Figure 7, in which DSC-Agent outperforms other methods by a large margin.

## 6 CONCLUSION

We have proposed a multi-stage learning framework for solving FPS games using general RL. The learning system consists of a number of advanced techniques and we have verified their importance by conducting comprehensive results in the experiments. The results demonstrate that our agent surpasses previous top ranking AIs by a large margin, by taking advantage of its diversified strategies instead of larger neural networks or hand-craft tricks. We also show that solving FPS games using general RL methods is possible without the usage of expert demonstrations and opponents' information. The proposed learning framework can be easily extended to other FPS games, and we are interested to apply this framework to solve real-world applications that share similarity with FPS games in future.

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

APPENDIX

## A   ENVIRONMENT

ViZDoom (Kempka et al., 2016) is an AI research platform based on the FPS game Doom. We adopt the CIG 2017 competition track 1 (Wydmuch et al., 2018), where 8 AI players join in a maze (shown in Figure 8) and play against each other. After a period of 10 minutes (in-game time), the players are ranked by the FRAG, which is defined as defeating count minus suicides (due to own rocket splash).

In our experiment, the observation is an RGB image, which is the first-person-view raw screen pixels as what a human player sees. The action is discrete in the size of 10, representing "TURN_LEFT", "TURN_RIGHT", "MOVE_RIGHT", "MOVE_LEFT", "MOVE_FORWARD", "MOVE_BACKWARD", "SPEED", "TURN180", "TURN_LEFT_RIGHT_DELTA", "ATTACK". It should be emphasized that, these buttons can be use at the same time, combine as the joint actions, to facilitate decoupling action and auxiliary action.

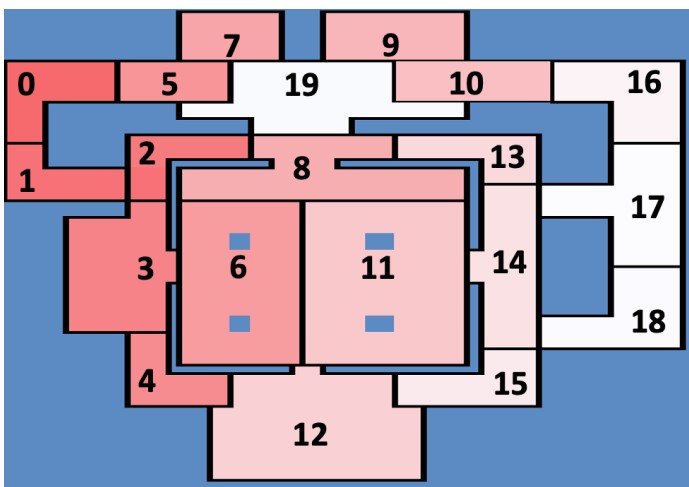

Figure 8: Map of ViZDoon track 1: we discretize the map into 20 areas, each of them is also used to represent an opponent with a fixed strategy style.

## B   TRAINING ARCHITECTURE

We used a multi-agent deep reinforcement learning framework designed for improve the efficiency of large-scale training, which achieves a high throughput and a reasonable scale-up when performing distributed training. Our architecture is based on an actor-learner structure (Espeholt et al., 2018): a large collection of over 1000 actor processes play ViZDoom deathmatch with prioritized fictitious self-play at the same time. There are 35 raw frames for one in-game, in our experiments we use a frame-skip = 2, henceforth the in game fps is 17.5fps. We use 1,152 CPU cores and 32 GPUs in toal, and the GPU type is M40 where only TCP connection is used for Horovod allreduce. After every 200 player time steps, the trajectory of experience from each player's point of view (observations, actions, rewards, infos) is sent to the learner.

## C   DSC-AGENT IN STAGE 2 AS OPPONENTS

The DSC-Agent in stage 2, which are trained on stage one and two, but without learning the diversity strategic control. The scope of their preferred activity distribution of 20 parts of the map as shown in Figure 8. It should be emphasized that, There are several noteworthy places, including No. 0 on the upper left side of the map, where the view is broad, and the terrain is flat; No. 3 is on the left side of the corridor; No. 6 is in the inner area of the map; No. 8 is the connecting area between the inner

and outer parts of the map; No. 12 Located on the lower side of the map, it is relatively hidden; No. 17 is on the right side of the corridor; No. 19 is on the upper platform of the map.

# D    SUPPLEMENTAL EXPERIMENT

Figure 9 show our result of experiment, in which we evaluate our DSC-Agent and other AI agents (Marvin, F1, Axon, and YanShi) fighting 7 opponents (DSC-Agent in stage 2) who all have fixed strategic strategies in one round. Figure 10 show our DSC-Agent' averaged distribution of strategy when fighting 7 opponents (DSC-Agent in stage 2).

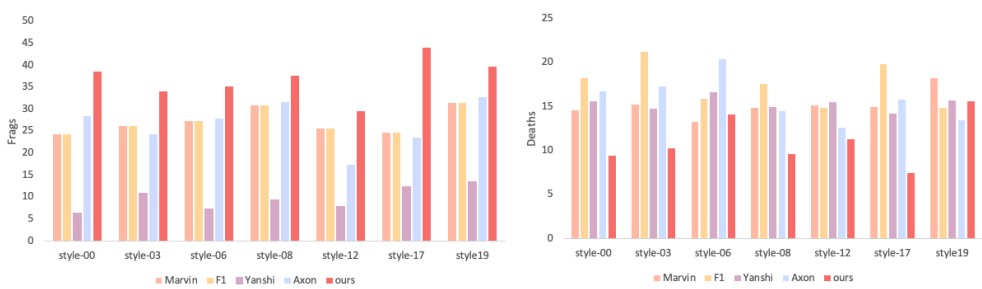

Figure 9: Average frags/deaths per AI against 7 opponents of fixed strategy style (10 episodes of 10 minutes each).

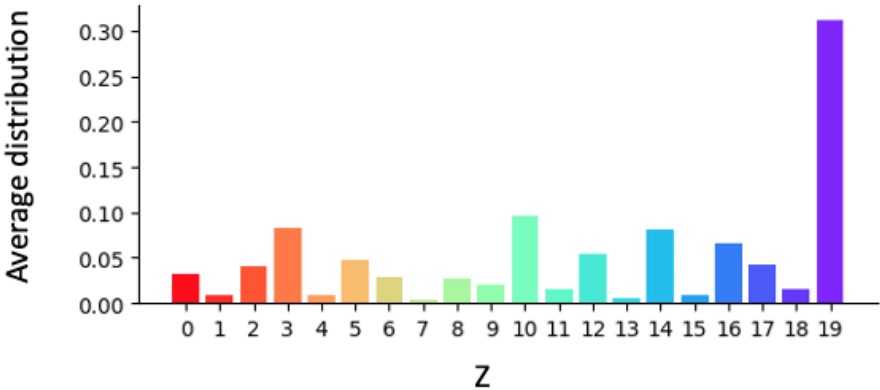

Figure 10: Averaged distribution of strategy: we carry out 10 rounds each type (the total is 20) of bots respectively, and calculated the average strategy distribution of DSC-Agent.

# E    NETWORK ARCHITECTURE

We employ a neural network shown in Figure 11 which consists of 3 blocks of convolution layer followed by 2 blocks of max pooling layer and an LSTM block. The inputs are the original observation and the enlarged observation in the center area. And the input z is from its own strategic head.

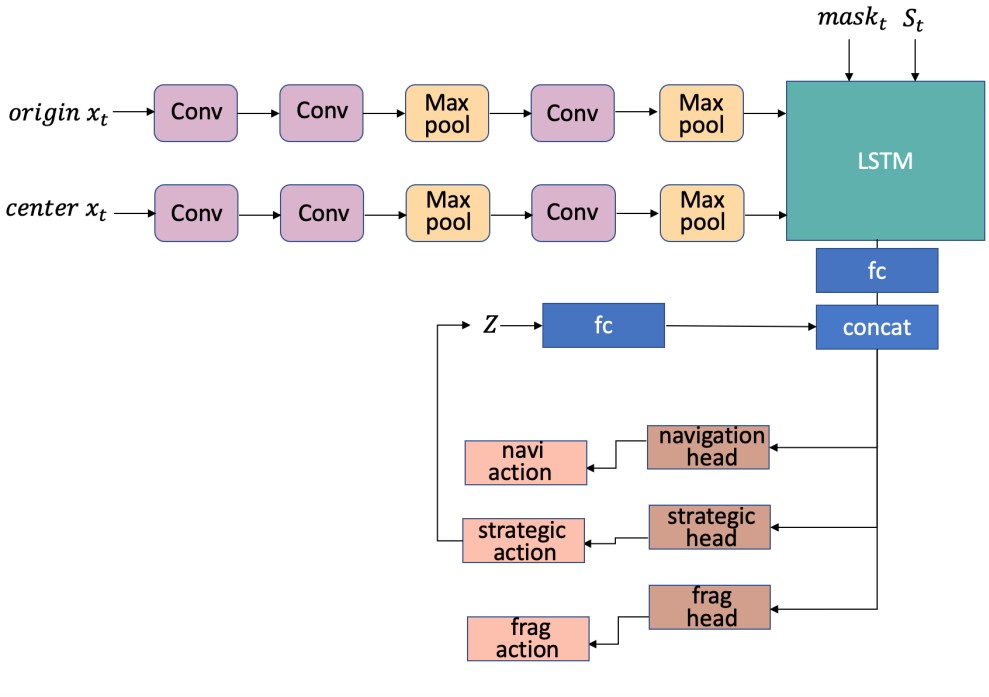

Figure 11: Overview of the network structure of DSC Agent.

