# OpenReview forum: "Superior Performance with Diversified Strategic Control in FPS Games Using General Reinforcement Learning"
_ICLR.cc/2022/Conference — ICLR 2022 Submitted_

### Official Review · Reviewer_jKT2 · 2021-10-29

**Correctness:** 3
**Technical Novelty And Significance:** 2
**Empirical Novelty And Significance:** 2
**Recommendation:** 3
**Confidence:** 4

**Main Review:**


Experiments are well-designed, and ablation studies are also presented. Visualizations make the paper much more interesting and understandable. This paper shows that instead of using larger NN, assembling existing techniques are also able to achieve SOTA performance.
Although the network is trained end-to-end,  they do use some hand-crafted domain specific rules in stage 2 and stage 3. Also, in this multi-stage learning system, each stage is manually selected/formed.

Reproducibility:

The neural network architecture is provided, but the used code and settings of hyper-parameters are not presented. 1152 CPUs and 32 GPUs are used, so I guess this work could be really hard to reproduce.


Relation to prior work:

Authors briefly explained the used related techniques and also baselines they used for comparison. Since the proposed learning framework is so different, there are not too much comparisons and relations depicted here. The proposed learning system is a multi-stage system, maybe it is better to cite some papers which are also using multi-stage learning system to solve tasks/games.


Could be improved:

Authors proposed a multi-stage learning system, and it would be interesting to compare with a flat learning system with the same/similar architecture;

Not only report means of different runs in Fig.6, but also report the standard error;

To make the loss terms more detailed, in Formular 2 and 3, losses for RGPS and policy distil are not specified;

The implementation work is impressive, but the description of the contribution and the method is not clear enough.

**Summary Of The Paper:**


The authors combined several different techniques from RL field to solve FPS game, ViZDoom. The game is splitted into three stages: navigation, frag and strategic control, respectively and the authors proposed a multi-stage learning system to tackle this problem. Navigation stage is solved by combining hindsight experience replay with PPO, which is named HPPO. Rule-guided policy search is used in frag stage and self-play is used for strategic control stage. Along the three stages, the trained model in the previous stage is always reused by adding new head to the neural network. The proposed multi-stage learning system outperforms SOTA agents(top ranking agents in open ViZDoom AI competitions) by a large margin. They also showed other advantages of the proposed system, such as learned skills are more diverse instead of fixed.



**Summary Of The Review:**


Strengths:

Experiments are well-designed, and ablation studies are also presented. Visualizations make the paper much more interesting and understandable. This paper shows that instead of using larger NN, assembling existing techniques are also able to achieve SOTA performance.



Weaknesses:

Although the network is trained end-to-end, but they do use some hand-crafted domain specific rules in stage 2 and stage 3. Also, in this multi-stage learning system, each stage is manually selected/formed.
Only a single game is used for empirical validation, the proposed method is quite complex, consisting of the combination of a number of existing methods,  the clarity of the contributed method is not sufficient, and no explanation is provided to explain how the method works and why the performance is better.
Overall the work is somewhat interesting, but not significant enough to warrant publication. Reproducibility is problematic.

---

> ### Author Response · Authors · 2021-11-23
> **Response to reviewer jKT2**
>
> ### "_**The neural network architecture ...hard to reproduce.**_"
> It is true that solving such a complex game requires large amount of computation resources. We will try to provide all the details to help the reuse of the techniques in other tasks.
> ### "_**Authors briefly explained ...system to solve tasks/games.**_"
> At present, we have not seen other similar learning framework in sovling general FPS tasks. The baselines compared adopts their specific training frameworks. After comparing with their work, the experiment verified DSC-Agent reached the SOTA of VDAIC Competition Track 1.
>
> We will reference the realted multi-stage learning approaches.
> ### "_**Authors proposed a multi-stage ...report the standard error.**_"
> Thank you very much for your suggestion. In fact, we have discovered that the agent trained by the flat learning system cannot even trained navi policy normally.
>
> These three stages are respectively used to train three types of agent control (navi control, frag control, strategic control). Many experiences have shown that multi-stage learning can effectively solve the difficulty of multi-task learning. If these three tasks are trained at the same time, the complexity of the action space will increase exponentially (imagine an agent learning from scratch, which needs to learn three tasks at the same time:
>
> 1. Learn to turn and move effectively according to the designated one hot code without stucking in the corner and hitting the wall.
>
> 2. Ensure that agent will not be attacked by itself and opponents while accurately attacking the opponents.
>
> 3. Quickly explore the entire map, find favorable regions to attack opponents, avoid their favorable regions, and defeat them).
>
> If any prerequisite task of agent is not satisfied in multi-task training, other tasks will quickly fall into local optimization and cannot be effectively explored anymore. In addition, multi-task learning also faces the problem of credit assignment (such as which control at which timestep should be assigned the reward after defeating a opponent) and different time scales (stage 3 is different from the first two stages).
>
> We have verified this conclusion in previous experiments, and we will report them in a later version of the paper.
>
> ### "_**To make the loss terms more detailed ...distill are not specified.**_"
> We will report standard error and Formula 2 and 3 in the future version of the paper
> ### "_**The implementation work is impressive ...not clear enough.**_"
> Thanks for your positive comment on our work. Please refer to Question 4 in Reviewer3
> ### "_**Although the network is trained ...is manually selected/formed.**_"
> **We have to clarify that the hand-crafted rules in the second and third stages are embedded in neural networks in the way of distillation, instead of applying the rules explicitly**. So, these rules are not persistently used, and they can be removed after the agent can explore reasonable states itself. The RGPS technique was first introduced in [1]. Similar to the pre-training of human expert strategy, the purpose of RGPS is to improve the exploration efficiency for frag / strategic level control. **This is necessary in many previous work, such as AlphaStar[2], and Marvin (the champion in Track1 of ViZDoom AI Competition 2017)[3]**.
>
> For the multi-stage learning framework, the tasks in the stages are different from each other, and these three tasks are **_ordinally correlated_**. The learning of strategic control is based on the ability to perform navi control and frag control, otherwise the agent will not be able to get the rewards for accurately going to the designated goal and defeating the opponent. The learning of frag control is also based on navi control, otherwise the reward for attacking opponents during normal movement cannot be obtained. Therefore, training in multi-stage is a natural way of sovling the FPS games.
> ### "_**Only a single game is used for ...Reproducibility is problematic.**_"
> It should be emphasized that our work aims to solve VDAIC Competition Track 1. Many previous work also used complex engineering techniques to solve this challenge. In addition, this challenge is the same as other FPS game tasks, so the method we propose can be applied to any similar FPS game, giving a standard technical framework for FPS game training. In the later version of the paper, we will report the implementation details of the parameters in detail, and supplement experiments to verify the role of each component in improving performance, especially the strategic control.
>
> [1]Han, Lei, et al. Tstarbot-x: An open-sourced and comprehensive study for efficient league training in starcraft ii full game. arXiv preprint arXiv:2011.13729 (2020).
>
> [2]Vinyals, O, et al. Grandmaster level in StarCraft II using multi-agent reinforcement learning. Nature 2019
>
> [3]Wydmuch, et al. Vizdoom competitions: Playing doom from pixels." IEEE Transactions on Games 2018

---

### Official Review · Reviewer_ahya · 2021-10-31

**Correctness:** 3
**Technical Novelty And Significance:** 1
**Empirical Novelty And Significance:** 2
**Recommendation:** 3
**Confidence:** 5

**Main Review:**

### Pros
- This work takes a problem-driven approach to improve the performance of a reinforcement learning agent on VizDoom. Although most of the RL literature starts from conceptual ideas to experimental results, I consider the present work to be valuable as it assembles and confronts some well-known techniques in RL to concrete problems.
- The impact of the action wrapper and the rule-guided policy search is a great example that sometimes less is more. Using simple tricks on top of generic methods can sometimes be much more effective than complicated approaches.

### Cons
- The quality and clarity of the writing is problematic. Besides the numerous spelling mistakes, the imprecise language often confuses the reader. For instance:
	* "...the state transition is not static", we generally talk about stationary or non-stationary environments. I'm not sure what static means in this case?
	* "..., while they are enabled to act diversely when the environment dynamics changes." The term `diverse` in the context of reinforcement learning is most often associated with Quality-Diversity that aims to generate a large collection of diverse solutions/policies. It is unclear whether the authors want to highlight the ability of their agent to adapt to changing dynamics (few-shot learning for instance) or if they want their agent to learn a policy performing well against opponents displaying a diverse set of strategies. For this reason, the method name, Diversified Strategic Control, is confusing as well.
	* The title mentions "General Reinforcement Learning". I'm not sure what the authors try to communicate? Besides the fact that general reinforcement learning feels undefined, the described approach uses tricks specific to FPS, like the statistical map provided as extra features to the agent's network or the rules defined in the 2 RGPS losses.
- As highlighted by the authors, the proposed approach consist of a combination of existing techniques. Although replicating and using these techniques require significant engineering effort, these can't be considered novel contributions. Other works that took a similar work setting, often made novel and original contributions, e.g. AlphaStar and their league system.
- The work is certainly complex from an engineering point of view: handcrafted curriculum learning, hindsight experience replay, distributed training strategy. As such, particular attention to providing all the required information to replicate the work has to be initiated. This aspect of the work is deeply lacking.
	- The values of all the hyper-parameters are not provided e.g. optimizers, losses coefficients, training parameters, etc.
	- Many terms and aspects are not defined. The authors don't explain: how they applied Policy Distillation, how the prioritized self-play has been implemented and put in action, what's the training dynamic (how long each stage lasts), where do the baselines come from (were they re-implemented or available as part of the competition?), etc.
	These missing bits of information would make the replication of the results nearly impossible to achieve and don't help the reader understand how the approach fully works and how it was implemented. I would encourage the authors to make a few steps in this direction as the lack of transparency on their method is puzzling.

### Questions
- Figure 5 only provides the score for a few goals. Why is that, and could the performance of the agents for the other goals be provided in the appendix?
- Regarding the training at stage 3, the authors mention that the selection of goals should be infrequent to avoid goal switching. Have the authors tried to constrain the switching of goals by restricting the goals to be adjacent to the current agent's goal, or by introducing a penalty in the reward?
- How are the end-of-life events processed? When the agent dies, is it considered the end of the episode? What are the implications for the hidden state of the LSTM, does it persist over the 10 minutes of the Deathmatch?
- What are the motivations behind having chosen PPO? The work requires applying Hindsight experience replay and importance sampling which increases the variance of the gradient. Have the authors tried using an off-policy algorithm that suits more easily HER?
- I would like additional clarifications on the goal selection. If I understand correctly the third head can decide the goal. If, for instance, the agent detects an enemy in an area that is not its current goal. How can the agent decide to change its strategy given that the third head is only activated once the agent reaches its current goal?

### Minor Comments
- Hindsight Trust Region Policy Optimization appears twice in the bibliography.
- Figure 2 is really hard to understand. It would benefit from a more extensive caption describing in greater detail each component and the transitions from one stage to the next.


**Summary Of The Paper:**

This paper proposes an approach to improve the performance of reinforcement learning agents on the ViZDoom FPS game (Deathmatch mode as used in the AI competition 2016, 2017). The method consists of a combination of existing techniques, e.g. PPO, Hindishght Experience Replay, Rule-Guided Policy Search, (Prioritized) Self-Play. The resulting algorithm is compared to the top-ranked agents from previous ViZDom AI Competitions (2016, 2017). The approach is shown to outperform these baselines and an ablation study is provided to highlight the impact on the performance due to the integration of each system component.

**Summary Of The Review:**

The paper improves the performance over the previously top-ranked solutions of the ViZDoom competition. Their approach consists of a combination of known technics well-engineered together. As such, the work lacks originality and novel contributions. In addition, many pieces of information that would enable a complete understanding of the method and the reproducibility of the results are missing. Consequently, I consider that the current version of the work does not meet the required standard for acceptance.

---

> ### Author Response · Authors · 2021-11-23
> **Response to reviewer ahya**
>
> Thank you very much for some guidance on our writing, it does create some difficulties for the reader.
> ### "_**'the state transition is not static' ...in this case?**_"
> Thank you for your advice. What we want to express here is that in a muti-agent task like VDAIC Competition Track 1 is a non-stationary environment, which will be corrected in the later version of the paper.
>
> ### "_**'while they are enabled to act diversely ... is confusing as well.**_"
>
> What we want to emphasize here is that DSC-Agent adopts different strategies to cope with opponents of different styles. Such strategies are generated by Diversified Strategic Control (DSC). Maybe we should consider a better terminology than the Diversified Strategy Control. Multiple Strategic Control can be a candidate.
>
> ### "_**The title mentions 'General Reinforcement Learning' ...in the 2 RGPS losses.**_"
> The reason why we use the “General Reinforcement Learning” has been mentioned in Section 1. This is to illustrate that the method we propose includes a number of general techniques in RL, including hindsight experience replay (HER) , hindsight proximal policy optimization (HPPO), Rule-guided Policy Search (RGPS), Prioritized Fictitious self-Play (PFSP). We will discuss in greater details of the Statistical Map and the Rules defined in the 2 RGPS losses in the updated version of this paper.
>
> ### "_**As highlighted by the authors ...e.g. AlphaStar and their league system.**_"
> we should emphasize that playing FPS game is complex, as it involves **imperfect information**, **first-person view**, **end-to-end training**, **multi-player interaction** and **multi-tasking**. A learning agent must overcome the difficulties including partial observation, large state and action space and unknown opponent strategies. This work discusses **how to address these difficulties and obtain a practical AI agent achieving state-of-the-art performance in VizDoom challenge**. We summarize the contribution of this work as follows. The novelty lies in:
>
> 1. Propose the Diversified Strategy Control for the FPS game.
>
> 2. Propose a goal conditioned variant of the PPO algorithm.
>
> 3. Design a framework for FPS training that combines the aforementioned methods as well as contemporary RL techniques.
>
> The empirical results we obtained are strong: we achieve the SOTA of VDAIC Competition Track 1.
>
> The proposed method can be potentially used in other related MARL scenarios, as it is a principled solution to a learning agent who needs navigate and perform other tasks when facing unknown opponents. For example, a sweeping robot has to avoid static/mobile obstacles and collect as much trash as possible in the target area.
>
> ### "_**The work is certainly complex from an engineering ...on their method is puzzling.**_"
> We will report parameters and implementation details and open source in a later version of the paper.
> ### "_**Figure 5 only provides ...in the appendix?**_"
> The left picture of Fig. 5 shows the most representative 6 types of bots with embeded goals. We will complement all the experiments in the revision.
> ### "_**Regarding the training at stage 3 ...a penalty in the reward?**_"
> As we mentioned in Section 4.3, only when the agent reached the specified region or reborn would it select next new goal.
> ### "_**How are the end-of-life ...of the Deathmatch?**_"
> The episode will only end when the 10-minutes game is over, and end-of-life events will not affect the LSTM. The LSTM's hidden state will be initialized only at the beginning of each game.
> ### "_**What are the motivations ...more easily HER**_"
> Please refer to the response to Question1 of Reviewer 2.
> ### "_**I would like additional clarifications ...I would like additional clarifications.**_"
> The agent cannot change its current goal when it detects an enemy in some other area before it achieves the current goal. However, this is not conflict with that the agent can automatically counter strike the detected agent, and after that it continues to its current goal. This implies that adaptively fighting with enemies with any taken goals should be learned and viewed as a part of the current goal-condition policy. Indeed, we have observed such instances in our evaluation games frequently.

---

### Official Review · Reviewer_73Nb · 2021-11-01

**Correctness:** 2
**Technical Novelty And Significance:** 2
**Empirical Novelty And Significance:** 2
**Recommendation:** 3
**Confidence:** 3

**Main Review:**

Strengths:
+ The paper is well-written and the logic is clear.
+ The visualization in the experiment clearly illustrates that the agent learns how to navigate, shoot, and other strategies.
+ The paper provides a careful design for the FPS game and achieves superior performance.

Weaknesses:
- To use off-policy data in hindsight experience replay, the paper adds an importance-sampling term to the PPO loss function, following the Hindsight TRPO method. However, there are no more techniques to reduce the high variance brought by cumulative multiplication terms. And the paper does not compare the HPPO with other goal-conditioned approaches in the experiment.
- I'm wondering why choosing different regions can represent different strategy styles. In my opinion, there are many policy styles in FPS games, such as aggressive/defensive/wandering, etc. These policy styles cannot be simply concluded by where the agent goes. For example, an agent can go to a same region for many reasons. It may go there to attack, to defend, or to detect the situation......
- In Stage 3, the strategic policy takes the statistical map recording the Frag scores as an input. I'm wondering how the agent cumulates Frag scores in unseen areas when testing, and whether the auxiliary network‘s prediction is accurate when facing unknown opponents.
- The experimental setup in evaluating strategy diversity is unfair. The evaluation opponent(Goal-00 to Goal-19) is used when training the DSC-Agent in stage 3. It is normal that DSC-Agent can get higher scores than other benchmark Agents.
- The paper should conduct more experiments to evaluate the strategic diversity of the agent which is the main contribution of the paper. How the agent dynamically changes its strategy as the game progresses, gathering more information about opponents. How does the agent develop strategies when fighting against agents not seen in training?

**Summary Of The Paper:**

This paper proposes a multi-stage learning framework for training high-performance agents in FPS games. Its agent can dynamically adjust its strategy according to different opponents, which is simply controlled by selecting different target areas. The paper propose a methodology called Hindsight PPO to solve goal-conditioned RL, though it may have some concerns. The framework combines several existing techniques such as rule-guided policy search,  action wrapper, prioritized fictitious self-play, etc. Experiments show that the framework learns an agent that can surpass previous top agents.

**Summary Of The Review:**

I think this paper deserves credit for training a superior performance agent by designing a multi-stage learning framework combining several techniques. However, this paper has some flaws:
- The proposed HPPO algorithm may suffer from high variance, and it is better to compare it with other methods.
- The relationship between strategy style and the target area is unclear.
- Some methods in Stage 3 are unclear.
- The experimental setup in evaluating strategy diversity is unfair.
- The paper should have more experiments to evaluate the strategic diversity of the agent.

---

> ### Author Response · Authors · 2021-11-23
> **Response to reviewer 73Nb**
>
> ### "_**To use off-policy data in hindsight experience replay ...approaches in the experiment**_"
>
> Learning with hindsight experiences with on-policy algorithms were studied in eariler approaches such as Hindsight policy gradients (HPG) [1] and Hindsight Trust Region Policy Optimization (HTRPO)[2]. HPG and HTRPO may suffer from high variance from the cumulative multiplication, and this is the reason that we propose HPPO to **apply clipped version of HTRPO to reduce the impact of high variance**. Even so, experiments in HTRPO have proved that in other discrete control tasks except ViZDoom, the performance of HTRPO is far better than the HER algorithm based on DQN.
>
> In addition the actor-critic style algorithms have been demonstrated to be effective in applications of large and complex game AI systems, as you may find examples of OpenAI Five, AlphaGo, AlphaStar, etc., because the actor-critic architecture is easily to be distributed for large scale RL. HER with deep Q-learning and its extensions have been well studied in the community of goal-conditioned RL. However, these methods are hard to be parallelized. So, we have not witnessed any large systems solved with hindsight Q-learning.
>
> Also note that in previous VDAIC Competition Track 1, on-policy algorithms (A3C/PPO) have been widely adopted [3].
> Therefore, we do not think it is necessary to compare these algorithms in the scope of this paper.
>
>
> ### "_**I'm wondering why choosing different regions ...or to detect the situation**_"
>
> Qualitatively, the strategy in an FPS game can be addressed in several dimensions. One dimension can be those **“low-level”** skills like dodging, shooting, tracking and firing in advance. Another dimension can involve **“high-levels”** styles like aggressive/defensive/wandering. The dimension we choose is about “what regions to go”, called DSC-Agent in our paper. The DSC also characterizes high-level strategies and can imply other high-level strategies like aggressive/defensive/wandering.
>
> In regards of opponent modeling, the DSC-Agent **expresses the aggressive/defensive/wandering strategy depending on its current knowledge of the opponents seen so far by going to different regions**. For instance, if an opponent attacks well in some specific regions, the DSC-Agent will collect much lower scores in these regions and henceforth avoid such regions, showing the sign of defensive style. If the DSC-Agent explores the map and discover other regions where it easily detects and eliminates the opponents, the collected FRAG scores will be high. The DSC-Agent will thus show up more frequently in such regions, turning out to be an aggressive style. On each episode beginning, the FRAG scores are initialized to zero, and the DSC-Agent wander on across each region update the FRAG statistics.
>
> In regards of modeling the environment,  the DSC-Agent **expresses the aggressive/defensive/wandering strategy also need to take regions‘ properties into consideration**, which are different from each other in several aspects: the terrain (z-axis height), the obstacles pattern (which can be used for crouching and defending), the vision, the supplies (e.g., the MedKits and Magazines to pick up), etc. Therefore, each region can be more suitable for a specific strategy. The regions with many corners/slopes are good for defending, while the regions with broad view field are better for attacking (such as goal-19).
>
> In summary, the DSC-Agent characterizes each region, taking into account both the opponent behaviors (whether an opponent performs better/worse in the region) and the environment (the terrain, the supplies, etc.). This way, the DSC-Agent shows more abundant strategies.
>
> ### "_**In Stage 3, the strategic policy ...facing unknown opponents.**_"
>
> It should be emphasized that the VDAIC Competition Track 1 presents a fixed known map during both training and testing. In training, we are able to take the ground-truth coordinates of each agent so as to accurately count the FRAG scores in each region. However, it is disallowed to access the coordinate information during testing by the criterion of VDAIC Competition Track 1. We thus adopt a separate module, trained via supervised learning, to roughly estimate the coordinate by taking as input the observation history. During testing, this module predicts in what region the agent is, which permits the updating of the region-wise FRAG score counting.
>
> ### "_**The experimental setup in evaluating ...than other benchmark Agents.**_"
> ### "_**The paper should conduct more experiments ...when fighting against agents not seen in training?**_"
>
> We will add more experimental details to verify our conclusions, thank you very much.
>
> [1]Rauber, Paulo, et al. "Hindsight policy gradients." ICLR 2019
>
> [2]Zhang, Hanbo, et al. "Hindsight Trust Region Policy Optimization." IJCAI 2021
>
> [3]Wydmuch M, et al. "Vizdoom competitions: Playing doom from pixels". IEEE Transactions on Games, 2018

---

> > ### Comment · Reviewer_73Nb · 2021-11-23
> > **Feedback on authors' response**
> >
> > Thank you for your response.
> >
> > About Question 1:
> > I'm wondering how often the clipping operation occurs in HPPO?
> > If the frequency is high because of the cumulative multiplication, HPPO will not update most state-action pairs, suffering from high bias.
> > If the frequency is otherwise low, the cumulative multiplication is close to one. It means the agent will execute almost the same strategy under different goals, which shows the failure of the multi-goal learning.
> > Either situation shows that HPPO is problematic.
> >
> >
> > About Question 3:
> > The experiments are not clearly explained, and the results are not convincing. During training, opponents are agents with fixed targets. However, opponents' strategies are unknown in testing, and they do not have fixed targets anymore. How did the method deal with unknown opponents in testing? Whether the auxiliary network's prediction is accurate when facing unknown opponents?

---

### Official Review · Reviewer_9yoB · 2021-11-02

**Correctness:** 3
**Technical Novelty And Significance:** 3
**Empirical Novelty And Significance:** 3
**Recommendation:** 3
**Confidence:** 3

**Main Review:**

Strengths:
1. The overall performance seems to be better than the state-of-the-art.

2. The motivation of each component is intuitive and straightforward.

3. The presentation of this paper is clear.


Weaknesses:

1. The baselines compared are from 2016 & 2018. How're latest competitions or related work after them? It seems like this part of the related work is missing, while the following work can be easily found:
- Song, Shihong, et al. "Playing FPS Games With Environment-Aware Hierarchical Reinforcement Learning." IJCAI. 2019.
- Xu, Zhiwei, et al. "HAVEN: Hierarchical Cooperative Multi-Agent Reinforcement Learning with Dual Coordination Mechanism." arXiv preprint arXiv:2110.07246 (2021).

2. The ablation study in this paper is not convincing enough to show the usefulness of different components and different stages. For example, there is no clear evidence showing that the whole process has to be done in a three-stage way. How about merging two of them, or learning them in one step?

3. The source code of this paper is unavailable, which makes it hard to reproduce and understand the details in the training process.

**Summary Of The Paper:**

This paper proposed a multi-stage learning framework for solving FPS games, with hindsight experience replay, goal conditioned reinforcement learning, and prioritized self-play. This whole work includes an overall solution, with combinations of existing work.

**Summary Of The Review:**

With the above strengths and weaknesses, I tend to give "marginally below the acceptance threshold". But I'm open change my rating after seeing the rebuttal from the authors to clarify my concerns.

---

> ### Author Response · Authors · 2021-11-23
> **Response to reviewer 9yoB**
>
> Thanks very much for your encouragement and constructive suggestions.
>
> ### "_**The baselines compared are from 2016 & 2018 ...be easily found**_"
>
> The competition, called **Visual Doom AI Competition (VDAIC)**, was held at IEEE Computational Intellengence and Games (IEEE CIG) annually from 2016 to 2018. In both 2016 and 2017, there are two Competition Tracks. Our submitted paper focuses on the **Competition Track 1:"Limited deathmatch on a known map"**, which was clearly mentioned in Section 3.
> While in year 2018, the Compettion Track 1 was changed to "Pass as many levels as possible in a limited time", which the Song S. et al. paper involves. The Xu, Z. et al. paper discusses a general Hierarchical MARL method. The authors indeed carry out experiments on the ViZDoom environment, however, both of the two papers seem to be out the scope of our discussion.
>
> ### "_**The ablation study in this paper ...or learning them in one step?**_"
>
> In the literature, curriculum learning (see [1] and the champion of 2016 Compettion Track 1:F1 [2]) is shown to be useful when training to tackle complex task.
> The three stages adopted in our paper are Navigation, FRAG Control (Shooting Skill), and Strategic Control. Note that they become **gradually difficult**, in the sense that **the later stage depends on and benefits from previous stage**. When trying to learn the three things simultaneously in a single stage, the difficulty may inflate, and the learning turns out to be intractable. Suppose an agent that learns from scratch to play an FPS game. Suppose also that we are trying to learn **simultaneously** all the following tasks :
>
> 1) Turning and moving effectively according to the designated one hot coding, without getting stuck in the corner and touching the wall.
>
> 2) Attacking the opponent accurately, without hurting itself (due to the rocket blast) and committing suicide.
>
> 3) Quickly explore the whole map, finding out the favorable regions (e.g., better terrain, more MedKits and Magazines to pick up, safe area with no lava, etc.) where the agent can eliminate opponents more often and avoid being attacked by the opponents.
>
> In a Multi-Agent Reinforcement Learning from randomly initialized Neural Net, the whole training is ruined when any prerequisite skill fails. In this case we may observe an agent that only learns to wonder near its spawning point, showing no high-level skills at all.
>
> On the other hand, multi-task learning cam suffer from **credit assignment problem**. For example, which of the three tasks should be responsible for the reward of defeating an opponent? Finally, the required “time resolution” is different for various tasks. The task 3) needs a larger timescale than task 1) and 2).
>
> Moreover, the core techniques used in each stage have been decomposed and evaluated in the ablation studies. Merging any two of the stages will result in inferior performance, and we will report the results in a later version of the paper.
>
> ### "_**The source code of this paper ...in the training process.**_"
>
> We will report the parameters and implementation details in the updated version of the paper, within which we will also open-source the code.
>
> [1].Bengio, Yoshua, et al. "Curriculum learning." ICML. 2009.
>
> [2].Wu, Yuxin, and Yuandong Tian. "Training agent for first-person shooter game with actor-critic curriculum learning." ICLR. 2016.

---

> > ### Comment · Reviewer_9yoB · 2021-11-29
> > **Summary**
> >
> > Thanks for your response.
> >
> > > both of the two papers seem to be out of the scope of our discussion.
> >
> > I don't believe at least the Xu, Z. et al. paper is out of the scope. The authors should compare with them, as HRL is also one way to solve complex tasks.
> >
> >
> > > The three stages adopted in our paper are Navigation, FRAG Control (Shooting Skill), and Strategic Control. Note that they become gradually difficult, in the sense that the later stage depends on and benefits from the previous stage.
> >
> > First of all, I'm not convinced that the tasks are gradually difficult if there is no theoretical/ empirical judgment on this claim.  Secondly, the authors should address the question on why dividing the whole task into three, rather than one (the simply replied "Merging any two of the stages will result in inferior performance, and we will report the results in a later version of the paper."), or four (for example, Navigation, Shooting Skill, Dodging Skill, Strategic Control).
> >
> > Given the feedback from the authors and reviews from other reviewers, I'm lowering my score to 3.

---

### Decision · Program_Chairs · 2022-01-20

**Decision:**

Reject

**Comment:**

The authors propose a method for training agents in FPS games, and achieve good results in a VizDoom setting. The method combines a number of different components and ideas, and it is not clear which of these are crucial to the success. In particular, ablations of the method are missing, as well as more runs to test variability and diversity. In addition, the paper is not all that easy to read. Reviewers had a number of partly overlapping concerns, of which I've tried to distil the main ones above. While the empirical results are promising, it is clear that much more work is needed to distil this method into generalizable knowledge.